# Metabolic Syndrome and Chronic Disease Risk in South Asian Immigrants: A Review of Prevalence, Factors, and Interventions

**DOI:** 10.3390/healthcare11050720

**Published:** 2023-03-01

**Authors:** Meena Mahadevan, Mousumi Bose, Kelly M. Gawron, Renata Blumberg

**Affiliations:** 1Department of Nutrition and Food Studies, Montclair State University, Montclair, NJ 07043, USA; 2San Diego Post-Acute Center, El Cajon, CA 92020, USA

**Keywords:** metabolic syndrome, chronic disease, South Asian, immigrants, acculturation

## Abstract

South Asians (SAs) are among the fastest-growing ethnic groups in the U.S. Metabolic syndrome (MetS) is a condition that is characterized by multiple health factors that increase the risk for chronic diseases, such as cardiovascular disease (CVD) and diabetes. MetS prevalence among SA immigrants ranges from 27–47% in multiple cross-sectional studies using different diagnostic criteria, which is generally higher compared to other populations in the receiving country. Both genetic and environmental factors are attributed to this increased prevalence. Limited intervention studies have shown effective management of MetS conditions within the SA population. This review reports MetS prevalence in SAs residing in non-native countries, identifies contributing factors, and discusses ways to develop effective community-based strategies for health promotion targeting MetS among SA immigrants. There is a need for more consistently evaluated longitudinal studies to facilitate the development of directed public health policy and education to address chronic diseases in the SA immigrant community.

## 1. Introduction

Metabolic syndrome (MetS) is a cluster of chronic disease risk factors, including abdominal obesity, hypertension, dyslipidemia, and impaired glucose tolerance [1]. Because MetS is directly related to risk for both cardiovascular disease (CVD) and type 2 diabetes mellitus [2,3], it can incur a high cost to both individuals and society. Reports from the International Diabetes Federation (IDF) have estimated that, in 2021 alone, the global healthcare costs associated with diabetes were USD 966 billion and are expected to grow to a projected rate of USD 1054 billion by 2045 [4]. As such, MetS and its related chronic conditions are public health issues of global significance. While the prevalence of MetS across different ethnic groups varies widely [5], the IDF estimates that, from 2011–2016, the overall prevalence of MetS in the United States was 36.9% [6].

Currently, there are distinct criteria for MetS that have been established by several institutions (Table 1). The most widely used definitions have been developed by the World Health Organization (WHO); the National Cholesterol Education Program Adult Treatment Panel (NCEP ATP III); the American Health Association (AHA), in conjunction with the National Heart, Lung, and Blood Institute (NHLBI); and the IDF [7].

Recently, the IDF and the AHA/NHLBI released a joint statement unifying several definitions into one consensus set of criteria [8]. All of these definitions incorporate cutoff points for body mass index (BMI) and waist circumference (WC), blood glucose and lipid levels, and blood pressure. Additionally, the consensus definition uses ethnic-specific criteria for WC, including distinct cutoff points for the Asian population. The term “South Asian” (SA) is typically used to refer to individuals with ethnic origins in India, Pakistan, Bangladesh, Nepal, Bhutan, Maldives, or Sri Lanka. Over the last two centuries, individuals of South Asian (SA) origin, particularly those originating from India and Pakistan, have immigrated to different countries, including the U.S. According to census data, there were over 4.5 million SA immigrants living in the U.S. in 2022, making this the fastest growing ethnic group in the country [9].

Studies have reported disparities in chronic disease between SA immigrants and the general U.S. population. Despite a lower average BMI [10], SA immigrants have higher prevalence rates of both diabetes [11,12,13,14] and CVD [15,16] compared to the majority population. Individuals of SA descent show higher amounts of abdominal fat and stronger insulin resistance when compared to individuals of European descent [17,18,19], suggesting an increased risk for metabolic syndrome. However, there are few large-scale studies examining MetS in the SA immigrant population. The purpose of this review is to compile and summarize studies focusing on MetS in SAs residing in non-native countries and to explore the factors that may influence the high burden of chronic disease observed in this population. Additionally, this review seeks to identify existing interventions and make suggestions for further research and practice to more effectively address the chronic disease burden in this burgeoning, yet underserved community.

## 2. Materials and Methods

For the determination of MetS prevalence, factors, and interventions among SA immigrants, a literature search was performed with the PubMed (National Library of Medicine, Bethesda, MD, USA) search engine. Multiple PubMed literature searches were performed using the following search terms and combinations contained in all fields of publications: (“Metabolic Syndrome”), (“Metabolic Syndrome” AND “South Asians”), (“Metabolic Syndrome” AND “South Asian Immigrants”), (“Metabolic Syndrome” AND “India”), (“Metabolic Syndrome” AND “Pakistan”), (“Metabolic Syndrome” AND “Bangladesh”), (“Metabolic Syndrome” AND “Sri Lanka”), (“Metabolic Syndrome” AND “Nepal”), (“Metabolic Syndrome” AND “Bhutan”), and (“Metabolic Syndrome” AND “Maldives”). Additional searches were performed to obtain articles pertaining to factors associated with MetS, such as acculturation, diet, physical activity, genetic/biochemical, and other psychosocial and environmental factors (e.g., food access and food availability). Finally, searches were also performed to identify interventions and lifestyle approaches to treat and manage MetS in these groups. This review method allowed for an in-depth examination of articles (Figure 1). As the focus of this review was SA immigrants, exclusion criteria were studies that did not relate to any of the aforementioned countries of origin, or the prevalence, contributing factors, or interventions targeting MetS in this population.

## 3. Results

**The Prevalence of MetS in SAs residing in non-native countries:** Studies examining the prevalence of MetS in SAs residing in non-native countries have reported rates ranging from 20–51% (Table 2). A large 2005 study (*n* = 1603) conducted in the U.K. reported a prevalence of 31–46% among SA immigrants, which was significantly higher than that among European subjects (9–18% prevalence, *p <* 0.001) [20]. A subset of that population, consisting of only males (*n* = 1420), showed a MetS prevalence of 44.6%; a higher prevalence compared to Europeans was still observed [21]. Another U.K. study found that MetS prevalence in SAs (*n* = 245) was nearly double what was observed among white subjects (39% vs. 20%, *p <* 0.001) [22]. Smaller studies in the U.S. [23] and the Netherlands [24] reported similar findings. In contrast, one study in Canada found that there were no significant differences in MetS prevalence between SAs (25.9%, *n* = 342) and whites (22%, *n* = 326) [25]. The specific countries of origin were not described in this study.

Among the large-scale studies conducted in the U.S. (*n* = 1403), one study showed an overall prevalence of 47% in a population of Sas residing in the country [26]. An earlier study observing Sas in the U.S. (*n* = 1445) found the prevalence of MetS to be 27% [27], while another report found that, among a population of 997 SAs living in the U.S., the prevalence of MetS was 37.6% [28]. These, and other large-scale studies in the U.S., have been comprised of SAs mostly of Indian descent [29,30]. One study involving Bangladeshi men living in Texas, USA (*n* = 91) reported MetS prevalence at 38% using NHLBI/AHA guidelines [31], while another showed that the highest prevalence of MetS was among Bangladeshi participants [26]. This suggests that the degree of variability in prevalence reported in these studies may be related to the ethnicity or country of origin of the participants.

**Prevalence of MetS in SAs residing in native countries:** A few studies have examined the prevalence of MetS among SAs residing in their native countries [32,33,34,35,36,37,38] (Table 3). Researchers have suggested that the rising prevalence can be attributed to the tremendous economic growth brought on by rapid transitions in nutrition, culture, and the environment in their native countries, especially over the past three decades. For example, some studies show that increased access to processed and refined foods [39,40,41,42], a decline in the availability and accessibility of healthier food options [43], along with a concurrent increase in sedentary behavior [44], are the results of globalization and industrialization [45,46]. Ultimately, these factors have led to an imbalance in energy intake and expenditure among its citizens [47]. Other environmental factors, such as an increase in air pollution, which is described as having a negative impact on the pathways regulating macronutrient metabolism [48], and psychosocial factors, such as depression, anxiety, and a lack of social support [49], have also been implicated in the development of chronic disease risks in these groups. More research examining these factors is necessary to get a clearer picture of whether or not disparities in the prevalence of MetS are due to differences in ethnicity.

**Differences in MetS in SAs between native and non-native countries:** Few studies have compared differences in risks for chronic disease among SA immigrants and their contemporaries residing in their native countries. These studies have consistently reported a higher BMI and waist circumference among SA immigrants compared to their native counterparts [50,51,52,53,54]. A 2011 meta-analysis of 10 studies reported that indices of obesity were greater in migrant Indian populations compared to native Indians [55]. Other studies have noted a higher risk for insulin resistance, diabetes, CVD, dyslipidemia, and hypertension among SA immigrants compared to the non-native country’s majority population [56,57].

In South Asians, polymorphisms for the gene encoding Apolipoprotein A-I, a protein component of high-density lipoprotein (HDL) particles, were significantly associated with MetS as well as low HDL levels, suggesting that the racial disparity of MetS between SAs and other races may be due to genetic differences [58,59,60,61,62,63]. While genetic predisposition is noteworthy, it does not provide a complete picture of the factors that contribute to greater risk for MetS in SA immigrants and their descendants. The sections below describe these factors.

**Acculturation:** The migration process has been linked to the increased prevalence of chronic diseases among SAs in non-native countries, suggesting the strong role of acculturation [64]. Acculturation is defined as the process by which a particular culture adopts the tenets and behaviors of a new culture, typically due to immigration [65]. Studies of immigrant cultures in the U.S. show that acculturation and its multiple dimensions might play a significant role in the development of chronic disease [66,67,68,69,70,71].

**Acculturation and diet:** Dietary acculturation refers to the process by which immigrants adopt the food habits of their host culture over a period of time [65]. Dietary behavior and the duration of residence in the U.S. have been identified as strong predictors of MetS in some SA immigrants [72]. For instance, several studies show that the adoption of a predominantly western diet, which is typically positively associated with increased duration of residence, contributes to the increased chronic disease burden in this group [73,74,75,76,77,78,79,80,81]. In one study, a high protein intake was related to an increased risk for diabetes, increased BMI, and higher waist circumference among SA immigrants living in America [82]. Of the total 146 participants in this study, 85 reported eating meat, while the rest adhered to a lacto-vegetarian diet (*n* = 29), or a lacto-ovo-vegetarian diet (*n* = 32). Among the meat eaters, the source of protein intake (vegetable, fish, or animal) was not associated with diabetes status. Compared to the meat eaters, however, both of the vegetarian diets were associated with lower insulin resistance. Also, those maintaining one of the two vegetarian diets had resided in the U.S. for a shorter duration than those who ate meat. An earlier study among Sri Lankan and Pakistani immigrants living in Norway found similar results, although a better understanding of the Norwegian language was associated with lower fat consumption [83].

It is worth noting here that using BMI as a proxy for obesity has several limitations. Unlike bioelectrical impedance, which measures both body fat and muscle mass, BMI is based solely on height and weight. Therefore, it is considered an indirect measure of body fat [84]. Additionally, it fails to reflect the differences and changes in these two components that may occur with gender, age, or physical activity level [85]. Among athletes, for instance, a higher weight may be due to higher muscle mass, thus resulting in overestimation [86]. Finally, two people may have considerably different BMIs, despite having identical or nearly identical percentages of body fat [87]. For these reasons, the use of BMI as a measure of obesity can misclassify, resulting in bias when estimating effects related to obesity. Nevertheless, these findings warrant further research, as they suggest that a transition away from a plant-based diet may contribute to increased chronic disease risk.

**Acculturation and physical activity:** Within a small population of Indian immigrants in California, U.S., even a moderate amount of physical activity was correlated with a lower prevalence of MetS in men [29]. Another study reported that duration of exercise was a statistically accurate predictor of MetS in SA immigrant women, but not in men [88]. The decrease in physical activity was associated with increases in both anthropometric (BMI and waist circumference) and biochemical (glucose and triglycerides) measures of MetS risk in this study. Increased acculturation was significantly associated with vigorous and moderate leisure-time physical activity, but not with light physical activity among Indian immigrants living in Canada [89]. A study of Indian immigrants in New Zealand that used pedometer measurements also reported decreased steps with increased acculturation [90]. These studies emphasize the role of physical activity in possibly mitigating the risk of MetS. However, the conflicting results with respect to gender and exercise intensity warrant further investigation.

**Interventions for MetS:** Several researchers have demonstrated the effectiveness of diet- and/or physical activity-based interventions in reducing the prevalence of MetS and chronic disease risks among various SA immigrant groups [91,92,93,94,95,96]. A randomized, controlled trial involving Pakistani immigrant men living in Norway (*n* = 150) found that after a 5-month intervention promoting physical activity (both cardiorespiratory and strength training), there was a slight decrease in the prevalence of MetS, and a significantly greater reduction in waist circumference and serum insulin concentration in the intervention group compared to the control group [97]. Another randomized, controlled trial involving Pakistani immigrant women living in Norway (*n* = 198) found that a 7-month intervention promoting both diet and physical activity resulted in a significant reduction in risk for Type 2 diabetes and MetS [98]. A smaller study of Pakistani immigrant women living in Australia (*n* = 40) also found that a 24-week intervention promoting healthy dietary behaviors and regular physical activity significantly decreased the participants’ BMI, blood pressure, cholesterol, and glucose levels compared to baseline [99].

Studies involving Indian immigrants have utilized a wide range of intervention strategies to address MetS, including nutrition education sessions on incorporating more plant-based protein sources, such as nuts [100,101]; exercise [102]; or a combination of diet- and exercise-based approaches [103]. Some of these interventions were delivered in a variety of formats, including one-on-one counseling sessions; group education workshops involving friends, peers, or family members; offered in the privacy of a participant’s home; or offered in a faith-based setting, such as a Hindu temple. Some educational components were also tailored to be more culture- and/or gender-specific, while others were tailored for audiences with low levels of literacy. Regardless of the approach, there were improvements noted across a range of parameters, including BMI, blood pressure, blood glucose, and waist circumference [104,105,106,107,108,109].

A review of randomized control trials from India showed that, in addition to diet, incorporating regular physical activity resulted in marked changes in risk factors for MetS, including a decreased BMI, waist circumference, and serum triglycerides and increased HDL-c levels [110,111,112,113]. Literature examining the impact of such interventions on MetS among other SA groups in their native countries is limited. Nevertheless, these data show that lifestyle interventions have the potential to positively impact MetS among SAs regardless of their country of residence.

## 4. Discussion

**Gaps in prevalence:** The harmonized consensus definition for MetS has recommended that waist circumference cutoff in Asian Americans be set at ≥90 cm for men, and ≥80 cm for women as a risk factor for MetS, which is in contrast to the values established by the WHO, NCEP ATP III, and the AHA/NHLBI, which are set to be higher at ≥102 cm for men and ≥88 cm for women, regardless of ethnicity [114]. These differences in criteria make it difficult to compare the prevalence of MetS across studies using different MetS definitions; this may explain the variability in these reports among SAs. In studies that describe MetS in SAs across multiple MetS definitions, disparate prevalence proportions have been reported within the same population [26,104,115]. Therefore, the discrepancies between MetS definitions need to be considered when comparing the prevalence of MetS in SAs across different studies. Additionally, most studies covered in this review are cross-sectional, suggesting a need for more longitudinal, prospective studies of MetS in SAs to develop more ethnic-specific criteria and apply these criteria in the formulation of effective treatment and prevention strategies for chronic disease in SAs. Future studies on MetS in the SA population warrant the application of a consistent MetS definition that takes into consideration ethnic-specific criteria, such as that of waist circumference in the IDF and harmonized consensus definitions.

**Gaps in interventions:** The studies reviewed thus far demonstrate that lifestyle interventions focusing on healthy eating and/or physical activity may be effective in preventing and treating MetS, in general. A scoping review comparing physical activity levels and their impact on MetS among Indian, Pakistani, and Bangladeshi immigrants settled in the UK describes these differences [116]. While levels of physical activity were generally lower compared to the majority population among all three groups, Bangladeshis had the lowest levels of physical activity, while Indians had the highest. In all three groups, women were less physically active than men, and older adults were the least physically active. Another review on cardio-metabolic risk factors among SA labor migrants who were hired for semi-skilled or unskilled jobs in the Middle East reported a high prevalence of being overweight/obese and related chronic diseases, such as diabetes and hypertension. Despite the high burden, there was a lack of focus on screening and inadequate provisions for health care [117].

These data show that educational components within interventions that are targeted toward reducing the rising burden of MetS in SAs need to vary depending on the group’s unique contextual needs; universal approaches that fail to consider ethnic and other societal inequities may fail in their compliance and overall effectiveness, likely widening the racial disparities in MetS prevalence. Most studies have grouped all SA immigrants together, even though there may be considerable differences in each ethnic group’s nutrition and lifestyle behaviors. Furthermore, the majority of the participants in these studies have been Indians. This group tends to be over-represented in South Asian chronic disease research.

Some studies have alluded to an inverse relationship between socioeconomic status and chronic disease in SAs living in non-native countries [26,31,70,76,85]. Studies on the relationship between mental health and chronic disease in this population have been inconclusive [30,70]. The influence of factors, such as health care access and differences in chronic disease risks due to occupational category or ethnicity, have not been studied sufficiently in this group either. Published literature that recognizes these differences and has reported the results separately is either limited or outdated [118,119].

**Implications:** Adopting a lower BMI cut-off for SA immigrants would serve to increase opportunities for improved diagnosis and intervention. However, a cross-sectional survey of primary care physicians practicing in a major southern city in the U.S. found only 9% of physicians reported measuring waist circumference, and only 21% of physicians were aware of ethnicity-specific guidelines. Most lacked the knowledge and training to appropriately assess overweight/obesity and related chronic disease risks in SA immigrants [120]. These data highlight the need for more culturally sensitive clinical strategies to reduce the burden of MetS in SA immigrant communities across the U.S. [121,122,123].

The socio-ecological model has been recognized as a systematic and coordinated approach to understanding and reducing disease risks, particularly among underserved and vulnerable population groups [124]. The model assumes that individuals are more likely to sustain disease treatment and management requirements within a comprehensive network that considers their individual (demographic characteristics, knowledge, attitudes, and beliefs), interpersonal (social support and size of social networks), environmental (availability of healthy food options and other resources), and institutional (public health policies) needs [125]. There appears to be a scientific gap, however, in developing and testing relevant policies and programs for SA immigrants that are designed around the constructs of the socio-ecological model. While most healthcare organizations provide basic services, they are largely based on models that focus on a single disease risk and do not address the personal, familial, or social needs of the various ethnic groups within this community. Few programs have addressed cultural relevancy in screening, diagnostic, and treatment tools. As summarized in Figure 2, translational and prospective research that provides a clear evaluation of the importance of ethnic-specific guidelines for diet and physical activity may ultimately help reduce the chronic disease burden in SA immigrants.

An optimal case management model for SA immigrants might be one that includes components that are not only culturally competent but ones that also strengthen and facilitate an individual’s environmental, psychological, and social networks [126]. For example, economic and zoning policies that ensure the availability of healthy foods and affordable preventive services in the neighborhoods in which these immigrant communities live and congregate may decrease risks for chronic diseases in the long term. Components that focus on personal control and self-esteem can help an individual regain better control over his or her health and attain weight management goals more effectively. Service facilities consisting of support staff that are more vigilant, especially to immigrant clients’ needs, may be critical to removing institutional barriers and building on assets to reduce the prevalence of MetS in this vulnerable population.

## 5. Conclusions

The toll of MetS and related chronic disease risks among individuals of South Asian origin is a rising public health problem. There is a need for an innovative treatment and management approach that balances ethnic-specific guidelines and recommendations with cultural and social norms and preferences, disease severity, disparities in access to neighborhood resources, and social support networks. Such an approach might affect SA immigrants to better engage in healthy behaviors.

## Figures and Tables

**Figure 1 healthcare-11-00720-f001:**
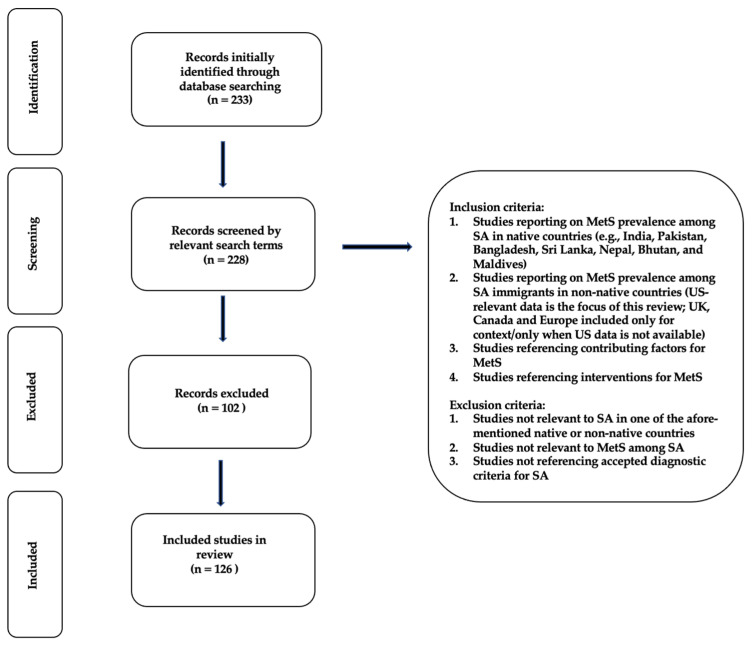
Prisma-flow diagram for literature review.

**Figure 2 healthcare-11-00720-f002:**
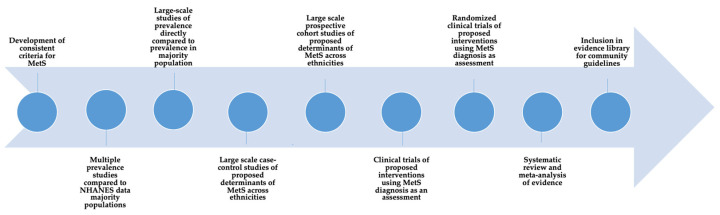
Conceptual model for data gathering for MetS and chronic disease risks in South Asian immigrants.

**Table 1 healthcare-11-00720-t001:** Major definitions/criteria for metabolic syndrome.

	Institution, Year					
	Consensus definition, developed by IDF and AHA/NHLBI, 2009	IDF, 2005	AHA/NHLBI, 2004	NCEP ATP III, 2001	EGIR, 1999	WHO, 1998
BMI	N/A	≥30 kg/m^2^	N/A	≥25 kg/m^2^	N/A	≥30 kg/m^2^
Central Obesity	WC cutoffs specific to the ethnicity being developed, currently using IDF criteria	WC for European: ≥94 cm (men) ≥80 cm (women) WC for Asian: ≥90 cm (men) ≥80 cm (women)	WC for all: ≥102 cm (men) ≥88 cm (women)	WC for all: ≥102 cm (men) ≥88 cm (women)	WC for all: ≥94 cm (men) ≥80 cm (women)	WHR ≥0.90 (men) ≥0.85 (women)
Fasting Glucose	≥100 mg/dL	≥100 mg/dL	≥100 mg/dL	≥110 mg/dL	≥110 mg/dL	≥100 mg/dL (as criteria for IR)
TGL	≥150 mg/dL	≥150 mg/dL	≥150 mg/dL	≥150 mg/dL	≥150 mg/dL	≥150 mg/dL
HDL-C	<40 mg/dL (men) <50 mg/dL (women)	<40 mg/dL (men) <50 mg/dL (women)	<40 mg/dL (men) <50 mg/dL (women)	<40 mg/dL (men) <50 mg/dL (women)	<39 mg/dL	<40 mg/dL (men) <50 mg/dL (women)
Blood Pressure	≥130/85 mm Hg	≥130/85 mm Hg	≥130/85 mm Hg	≥130/85 mm Hg	≥140/90 mm Hg, or hypertensive drug usage	≥140/90 mm Hg

Abbreviations: IDF, International Diabetes Foundation; AHA, American Heart Association; NHLBI, National Heart, Lung, and Blood Institute; NCEP, National Cholesterol Education Program; ATP, Adult Treatment Panel; EGIR, European Group for the Study of Insulin Resistance; WHO, World Health Organization; BMI, body mass index; WC, waist circumference; WHR, waist–hip ratio; IR, insulin resistance; TGL, triglycerides; HDL-C, high-density lipoprotein cholesterol.

**Table 2 healthcare-11-00720-t002:** Prevalence of MetS in SA in non-native countries.

Author, Year (SA Group Studied)	Non-Native Country	Criteria Used	Prevalence
Khan et al., 2016 (Pakistan, India, Bangladesh, Nepal, Iran, Sri Lanka, Afghanistan, Bhutan)	US	Modified harmonized definition by IDF and NHLBI	47%; Highest prevalence among Bangladeshi men
Garduno-Diaz et al., 2013 (India, Pakistan)	UK	IDF	20%
Andersen et al., 2012 (Pakistan)	Norway	IDF	47–51%
Dodani et al., 2011 (India)	US	IDF, WHO, NCEP ATP III,	29.7% (IDF), 13.3% (WHO), 40% (NCEP ATP III)
Flowers et al., 2010 (India, Pakistan, Sri Lanka)	US	IDF	27%; Prevalence significantly higher in men (31%) than in women (17%)
Misra et al., 2010 (India)	US	IDF, NCEP ATP III	37.6% (IDF), 32.4% (NCEP ATP III); Prevalence significantly increased with age in women, but not with men
Telle-Hjellset et al., 2010 (Pakistan)	Norway	IDF	41%
Rianon et al., 2009 (Bangladesh)	US	Modified AHA/NHLBI	38%
Balusubramanyam et al., 2008 (India)	US	NCEP ATP III	32%; Prevalence was higher in the older population
Ajjan et al., 2007 (India, Pakistan, Bangladesh)	US	IDF	39%; Prevalence was significantly higher compared to those of Caucasian descent
Williams et al., 2007 (India, Pakistan, Bangladesh, Sri Lanka)	UK	NCEP ATP III	22.2%
Forouhi et al., 2006 (India, Pakistan, Bangladesh)	UK	IDF	44.6%; Prevalence was significantly higher in South Asians compared to Europeans
Tillin et al., 2005 (India, Pakistan, Bangladesh)	UK	WHO, NCEP ATP III	46% Male, 31% Female (WHO), 29% Male, 32% Female (NCEP ATP III); Prevalence, using both criteria, was significantly higher compared to Europeans
Misra et al., 2005 (India)	US	NCEP ATP III	33.9%
Anand et al., 2003 (Not specified)	Canada	NCEP ATP III	25.8%; Prevalence was significantly higher compared to Chinese immigrants, but not those of European descent

**Table 3 healthcare-11-00720-t003:** Prevalence of MetS in SAs in native countries.

Author, Year	Native Country	Criteria Used	Prevalence/Contributing Factors
Adil et al., 2023	Pakistan	NCE ATP III	28.8%
Sundarakumar et al., 2022	India	NCEP ATP III	46.2% (rural) 54.8% (urban)
Ali et al., 2020	Bangladesh	NCE ATP III	22%
Subramani et al., 2019	India	NCE ATP III, IDF	72.7% (NCEP ATP III) 50.2% (IDF)
DeSilva et al., 2019	Sri Lanka	IDF	47.2%
Mehata et al., 2018	Nepal	NCE ATP III, IDF	15% (NCEP ATP III) 16% (IDF)
Sinha et al., 2013	India	NCEP ATP III, IDF	29.6% (NCEP ATP III) 20.4% (IDF)

## Data Availability

Not applicable.

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
