# Peer review of "Metabolic Syndrome and Chronic Disease Risk in South Asian Immigrants: A Review of Prevalence, Factors, and Interventions"

_healthcare, 2023, doi:10.3390/healthcare11050720_

Round 1

Reviewer 1 Report

This is an important topic for discussion. How does the prevalence of MeS compare between native and non-native countries? Are the confounding variable between the two settings similar or different? A particular statement (ref 72) regarding increased protein intake related to increased risk of diabetes needs more explanation... animal protein? plant protein? Also, BMI is not an ideal measure of obesity for a number of reasons - need to add clarification on limitations of use of BMI. Does lifestyle intervention work the same way in terms of outcome between the native and non-native settings?

Author Response

Thank you so much for your very insightful comments to strengthen the quality of this paper. We very much appreciate it. Please find below our response to each of your comments:

Comment 1: How does the prevalence of MeS compare between native and non-native countries? Are the confounding variable between the two settings similar or different?

Response: This is an excellent point. We have reorganized the Results section by adding new sections to address your point. The new sections titled Differences in MetS in SA between native and non-native countries, and Acculturation explain the variables contributing to the differences. See lines 134-154. Additionally, we have provided more clarification to Table 2 by indicating which receiving/non-native country was included, and which SA group was studied in each study. Finally, we have included a new Table 3 to provide prevalence data for native countries.

Comment 2: A particular statement (ref 72) regarding increased protein intake related to increased risk of diabetes needs more explanation... animal protein? plant protein?

Response: Clarification has been provided with details with supporting literature. See lines 162-166. Please also note that due to new literature and data added to the revised manuscript, this reference is now newly numbered as Reference #82.

Comment 3: Also, BMI is not an ideal measure of obesity for a number of reasons - need to add clarification on limitations of use of BMI.

Response: A new paragraph has been added under the section Acculturation and Diet under Results. This paragraph explains the limitations of BMI, with supporting literature. See lines 172-182.

Comment 4: Does lifestyle intervention work the same way in terms of outcome between the native and non-native settings?

Response: A new paragraph has been added under the section Interventions for MetS under Results. This paragraph explains this point supporting literature. See lines 221-227.

Reviewer 2 Report

Comments : A well considered article and I note that you have highlighted the lack of data about immigrants coming from countries other than India

Can I suggest the authors comment in more detail regarding the following points 

1. The South Asian countries are diverse in terms of their socioeconomic status between countries and between communities within a country. Recent migrations have arguably involved more highly educated economic migrants from the originating country  whose CVS and MetS risk profiles may not reflect those of a broad based national study from their country of origin. 
Nevertheless a table showing the published prevalence of MetS in the South Asian countries of origin eg India, Pakistan, Bangladesh , Nepal and Maldives of the SA immigrants and some data about the prevalence of MetS in the majority population or national studies of the receiving country would be helpful for readers to understand the assertion by the authors that  : The differences in these reports may be related to ethnicity of participants between studies. The highest prevalence of MetS in the report by Khan et al. was among Bangladeshi participants, while individuals of Bangladeshi descent were not represented in many earlier reports.

2. The point about the difference in diagnostic criteria ( notably the cut off waist circumference of 102 cm in males and  88 cm in females in studies using the  AHA/NHLBI, 2004 and NCEP ATPIII, 2001criteria vs those using the IDF , EGIR and WHO waist circumference criteria of 94 cm in males and 80 cm in females ) being a possible explanation for  the wide range of estimates of prevalence of MetS between studies is well taken but the authors also clarified that . I think you could clarify  your abstract to read something along the lines of  "MetS prevalence among SA immigrants ranges from 27%-47% in multiple cross-sectional studies,  using different diagnostic criteria, which is generally higher compared to other populations in the receiving country . " 

3. Within receiving countries for South Asian immigrants, different waves of immigrants of the same ethnicity or country of origin coming in at different times ( eg a postwar era, or a boom in tech or research based jobs  or an influx of unskilled workers to fulfill certain labour needs ) will have both different characteristics and needs and implications about what would then be culturally appropriate intervations .  The aim of this article seems to be to draw attention to the need for recognition of diverse risk factors and diverse needs of the South Asian subpopulation in the USA and other receiving nations , so a bit more discussion of these issues may be in order

Author Response

Thank you so much for your very insightful comments to strengthen the quality of this paper. We very much appreciate it. Please find below our response to each of your comments:

Comment 1: The South Asian countries are diverse in terms of their socioeconomic status between countries and between communities within a country. Recent migrations have arguably involved more highly educated economic migrants from the originating country whose CVS and MetS risk profiles may not reflect those of a broad-based national study from their country of origin. Nevertheless, a table showing the published prevalence of MetS in the South Asian countries of origin eg India, Pakistan, Bangladesh, Nepal and Maldives of the SA immigrants and some data about the prevalence of MetS in the majority population or national studies of the receiving country would be helpful for readers to understand the assertion by the authors that “The differences in these reports may be related to ethnicity of participants between studies. The highest prevalence of MetS in the report by Khan et al. was among Bangladeshi participants, while individuals of Bangladeshi descent were not represented in many earlier reports.”

Response: This is an excellent point. We have reorganized the Results section by adding new sections to address your point. The new sections titled Differences in MetS in SA between native and non-native countries, and Acculturation explain the variables contributing to the differences. See lines 134-154. Additionally, we have provided more clarification to Table 2, indicating which receiving/non-native country was included, and which SA group was studied in each study. Finally, we have included a new Table 3 to provide prevalence data for native countries.

Comment 2. The point about the difference in diagnostic criteria (notably the cut off waist circumference of 102 cm in males and  88 cm in females in studies using the AHA/NHLBI, 2004 and NCEP ATPIII, 2001criteria vs those using the IDF, EGIR and WHO waist circumference criteria of 94 cm in males and 80 cm in females) being a possible explanation for  the wide range of estimates of prevalence of MetS between studies is well taken but the authors also clarified that. I think you could clarify your abstract to read something along the lines of “MetS prevalence among SA immigrants ranges from 27%-47% in multiple cross-sectional studies, using different diagnostic criteria, which is generally higher compared to other populations in the receiving country." 

Response: This suggestion is well taken. The sentence you recommended has been added to the Abstract. See lines 13-15.

Comment 3: Within receiving countries for South Asian immigrants, different waves of immigrants of the same ethnicity or country of origin coming in at different times (eg. a postwar era, or a boom in tech or research-based jobs or an influx of unskilled workers to fulfill certain labor needs) will have both different characteristics and needs and implications about what would then be culturally appropriate interventions.  The aim of this article seems to be to draw attention to the need for recognition of diverse risk factors and diverse needs of the South Asian subpopulation in the USA and other receiving nations, so a bit more discussion of these issues may be in order.

Response: This is an excellent point. We have done two things to address your suggestion:

1. The new section Prevalence of MetS in SA residing in native countries under Results provides an explanation for this with supporting literature. See lines 118-132.

2. We have also re-organized the Discussion section by adding in more literature under new section headings. For instance, the new sections titled Gap in prevalence research, Gap in intervention research, and Implications provides explanation for your suggestions, with supporting literature. See lines 228-314.

Reviewer 3 Report

This is a well-written review of metabolic syndrome and chronic disease risk in South Asian immigrants. As the authors commented, these are under-observed ethnic groups with their own characteristics and health needs, so it seems to me an interesting topic that should be published.

I respectfully submit here a few observations:

-Is it a good idea to include the word "review" in the title?

-Line 31: More recent data will most likely be available in the 2022 Diabetes Atlas.

-Line 43: Tables should have horizontal lines only (check template). I recommend removing the bold letters from the table headers and changing the layout of the table (cut-off points as rows and institutions (IDF etc.) as columns).

-Usually, what the authors wrote in "results", following methodologies such as PRISMA reviews, is included in the materials and methods section, in a figure where the total number of articles, discarded articles (and why they were discarded) etc. is given. And the results give an overview of the information found, which will later be discussed, as you did.

-Table 2. Same comment regarding table formatting. Add a column with the citations of each study, in MDPI format, because you put author and year, but in APA style.

-Line 174-177: I recommend to the authors that they do briefly discuss this very important factor for the development of MetS in SA populations. 

Author Response

Thank you so much for your very insightful comments to strengthen the quality of this paper. We very much appreciate it. Please find below our response to each of your comments:

Comment 1: Is it a good idea to include the word "review" in the title?

Response: We think this is an excellent idea because it tells the audience what the paper is all about, at first glance. We have revised the title.

Comment 2: Line 31: More recent data will most likely be available in the 2022 Diabetes Atlas.

Response: This is a great suggestion. We have revised the statement to include newer data. See lines 30-33 under Introduction. Line 56-58 also includes new data. Please note that these newer references have been added to the References list as well. See Reference #4 and #9.

Comment 3: Line 43: Tables should have horizontal lines only (check template). I recommend removing the bold letters from the table headers and changing the layout of the table (cut-off points as rows and institutions (IDF etc.) as columns).

Response: Changes have been made. See Table 1.

Comment 4: Usually, what the authors wrote in "results", following methodologies such as PRISMA reviews, is included in the materials and methods section, in a figure where the total number of articles, discarded articles (and why they were discarded) etc. is given. And the results give an overview of the information found, which will later be discussed, as you did.

Response: This suggestion is well taken. We have moved the sentences describing the method of search to Materials and Methods. See lines 72-89. We have also included a new figure (Figure 1) in this section. Additionally, to your point, the sections under the Discussion section in the original manuscript have all been moved to Results in this revised manuscript. See lines 94-227. Because of these changes, we have re-organized the Discussion section by adding new section headings as well as tightened the Conclusions section. We hope that these changes have helped re-organize the literature in a more meaningful way.

Comment 5: Table 2. Same comment regarding table formatting. Add a column with the citations of each study, in MDPI format, because you put author and year, but in APA style.

Response: Changes have been made with regards to horizontal lines. See Table 2. We checked with the journal editor Ms. Linn Sun, who advised us “You don't need to revise the reference
format. Later, our layout colleague will help you to revise to MDPI format.” Email dated 02/13/23.

Comment 6: Line 174-177: I recommend to the authors that they do briefly discuss this very important factor for the development of MetS in SA populations. “While not discussed in this review, the role of rapid industrialization and nutrition transition with respect to chronic disease in developing SA countries, particularly in comparison to the immigrant experience, needs to be further addressed.”

Response: This is an excellent point. We have done two things to address your suggestion:

  1. The new section Prevalence of MetS in SA residing in native countries under Results provides an explanation for this with supporting literature. See lines 118-132.
  2. We have also re-organized the Discussion section by adding in more literature under new section headings. For instance, the new sections titled Gap in prevalence, Gap in interventions, and Implications provides explanation for your suggestions, with supporting literature. See lines 228-314.

Round 2

Reviewer 1 Report

Authors have carefully considered the reviewer comments and provided clarification.